# TAMC: A deep-learning approach to predict motif-centric transcriptional factor binding activity based on ATAC-seq profile

**Tianqi Yang**[1,2]*, **Ricardo Henao**[3,4]*

**1** Department of Pharmacology and Cancer Biology, Duke University School of Medicine, Durham, North Carolina, United States of America, **2** Department of Cell Biology, Duke University School of Medicine, Durham, North Carolina, United States of America, **3** Center for Applied Genomics and Precision Medicine, Duke University School of Medicine, Durham, North Carolina, United States of America, **4** Department of Biostatistics and Informatics, Duke University, Durham, North Carolina, United States of America

* tianqi.yang@duke.edu (TY); ricardo.henao@duke.edu (RH)

## Abstract

Determining transcriptional factor binding sites (TFBSs) is critical for understanding the molecular mechanisms regulating gene expression in different biological conditions. Biological assays designed to directly mapping TFBSs require large sample size and intensive resources. As an alternative, ATAC-seq assay is simple to conduct and provides genomic cleavage profiles that contain rich information for imputing TFBSs indirectly. Previous footprint-based tools are inheritably limited by the accuracy of their bias correction algorithms and the efficiency of their feature extraction models. Here we introduce TAMC (**T**ranscriptional factor binding prediction from **A**TAC-seq profile at **M**otif-predicted binding sites using **C**onvolutional neural networks), a deep-learning approach for predicting motif-centric TF binding activity from paired-end ATAC-seq data. TAMC does not require bias correction during signal processing. By leveraging a one-dimensional convolutional neural network (1D-CNN) model, TAMC make predictions based on both footprint and non-footprint features at binding sites for each TF and outperforms existing footprinting tools in TFBS prediction particularly for ATAC-seq data with limited sequencing depth.

## Author summary

Applications of deep learning models are rapidly gaining popularity in recent biological studies because of their efficiency in analyzing non-linear patterns from feature-rich data. In this study, we developed a deep learning method to predict transcription factor binding sites based on chromatin accessibility profiles. Compared to previous methods using scoring functions and classical machine learning algorithms, our method forgoes the need for bias correction during signal processing and significantly increases the efficiency in extracting features at transcription factor binding sites. In addition, we showed that our method outperforms previous methods particularly for chromatin accessibility data with shallow sequencing depth. In this study, we applied our method to prediction of changes in binding sites of a transcription factor, CTCF, during early embryonic development

**Data Availability Statement:** Raw data access number and derived data supporting the findings of this study are listed within the manuscript and its Supporting Information files. Code for model

training and testing is openly available in TAMC Github repository at https://github.com/tianqiyy/TAMC.git.

**Funding:** The authors received no specific funding for this work.

**Competing interests:** The authors have declared that no competing interests exist.

based on bulk chromatin accessibility profiles. We then discussed about the potential application of our method to transcription factor binding site prediction using single-cell chromatin accessibility profiles as well as possible strategies to further improve the performance of our method in the future.

This is a *PLOS Computational Biology* Methods paper.

## Introduction

Transcription factors (TFs) are proteins that bind to conserved genomic sequence motifs and have functions in regulating gene expression [1]. Determining TF binding sites (TFBSs) is essential for deciphering the molecular mechanisms regulating gene expression across different biological conditions. Biological assays, such as ChIP-seq [2] and CUT&RUN [3], have been used as the standard experimental methods for mapping genome-wide interactions between TFs and chromatin. However, these experiments are resource-intensive and can measure only one TF at one time, which largely limits their applications in many situations. To address these limitations, computational methods, discussed below, have been proposed to impute TFBSs.

Traditional computational TFBS prediction methods have been using position weight matrices (PWMs) of TF binding motifs against DNA sequence to predict TFBSs [4,5], yet these methods suffer from high false-positive rates (FDR) [6]. Recent studies have shown that more than 90% of TF binding events take place at open chromatin regions [7] that could be mapped by enzymatic cleavage assays such as DNase-I sequencing (DNase-seq) [8] and Assay for Transposase Accessible Chromatin sequencing (ATAC-seq) [7]. Notably, bound TFs hinder the activity of cutting enzymes and leave footprint sites characterized with lower cleavage signal frequency comparing to surrounding regions [9]. Therefore, the TF-bound and unbound sites can be theoretically distinguished by their footprint pattern.

Several computational methods have been developed to investigate footprint patterns in chromatin cleavage profiles [10–19]. As chromatin cleavage events generated by cutting enzymes (e.g., DNase-I used in DNase-seq and Tn5 transposase used in ATAC-seq) are biased towards different sequences, previous tools initially designed for DNase-seq data often give poor predictions using ATAC-seq data [16,20,21]. By far, HINT-ATAC [16] and TOBIAS [17] are two representative footprinting tools specifically designed for ATAC-seq data, which has become the dominant data type for chromatin accessibility profile because of the simplicity of the assay itself. TOBIAS uses a simple footprint score (FPS) metric to characterize the footprint pattern at single-base resolution while HINT-ATAC uses a semi-supervised hidden Markov model (HMM) to predict footprint sites directly. Both tools significantly increase the accuracy in classifying bound/unbound sites from ATAC-seq data. On the other hand, the limitation is that they both require complex bias correction during signal processing because their models/algorithms are highly dependent on measurable footprint pattern to make predictions.

More recently, deep-learning models are rapidly gaining popularity in biological studies because of their efficiency in analyzing complex (non-linear) patterns from feature-rich data. Here, we introduced a new TFBS prediction tool named TAMC (**T**ranscriptional factor binding prediction from **A**TAC-seq profile at **M**otif-predicted binding sites using **C**onvolutional neural networks). TAMC takes advantage of signal processing strategies in HINT-ATAC and

TOBIAS except for the bias correction step to produce input signals that are then used to extract features of TFBSs using a 1D-convolutional neural network (1D-CNN) model (Fig 1A). By evaluating TAMC with different input signal configurations, we showed that TAMC does not require bias correction during signal processing and captures both footprint and non-footprint features of TFBSs efficiently. Importantly, TAMC models pretrained with multiple deeply sequenced ATAC-seq datasets significantly outperform HINT-ATAC and TOBIAS in TFBS prediction especially when using ATAC-seq data with limited sequencing depth. In our study, we have applied TAMC in predicting changes in binding sites of a specific TF, CTCF, during human zygotic genome activation (ZGA) using bulk ATAC-seq data. We also believe that TAMC is a competitive alternative method for TFBS prediction using single-cell ATAC-seq data.

## Results

### TAMC overview

TAMC takes a combination of footprint profile and genomic cleavage profile (signals and slopes) around 500bp from the center of potential TF binding sites predicted by their binding sequence motifs, or motif-predicted binding sites (MPBSs), as input signal (Fig 1B). The

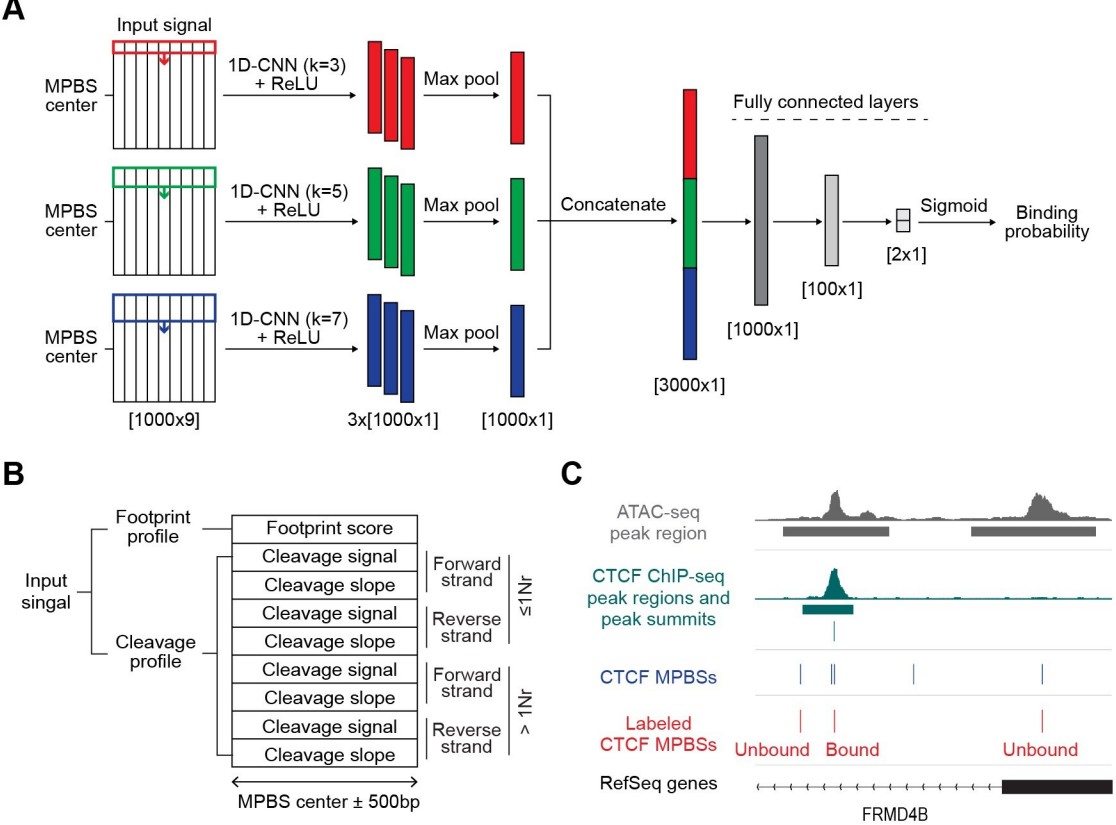

**Fig 1. TAMC model and input signal.** (**A**) Architecture of TAMC framework: three convolutional layers with 3 learnable filters of kernel sizes k = 3, 5, and 7, and ReLU activations, followed by max pooling, concatenation, and prediction of binding probability via a two fully connected layers with sigmoid activation. (**B**) Default TAMC input signal structure. >1Nr, ATAC-seq read fragments larger than one nucleosome size; ≤1Nr, ATAC-seq read fragments equal or smaller than one nucleosome size. (**C**) Representative tracks show strategy of labeling bound and unbound CTCF MPBSs in GM12878 cell type at FRMD4B gene locus. MPBS, motif-predicted binding sites.

footprint scores are calculated using the TOBIAS footprint scores metric, and the genomic cleavage signals and slopes are calculated and normalized using HINT-ATAC input signal processing scripts with modifications at the bias correction step. The genomic cleavage signals and slopes are further separated into 8 channels by strand and size of ATAC-seq reads. This results in each input signal has 9 channels in total– 1 channel for footprint profile and 8 channels for cleavage profiles (Fig 1B). The 9-channel input signals are fed into a 1D-CNN module and the convolution is performed with filter kernels moving at 1 base position (bp) per step in the direction from -500bp to +500bp from the motif center. As the length of binding sites varies between TFs, we utilized kernels with different sizes (k = 3, 5 and 7) to extract features for small and large binding sites (Fig 1A). For each kernel size, we repeated convolution for three times and the obtained three feature maps were max-pooled to extract the most salient feature at each bp within the input region (Fig 1A). The max-pooled feature maps originate from different kernel sizes are then concatenated before being fed to the fully connected layers and a final sigmoid activation function to make predictions of TF binding probability (Fig 1A).

The TAMC framework was trained and tested using published paired-end ATAC-seq data of three human cell types (GM12878, HepG2 and K562) (S1 Table). ChIP-seq data obtained from the same cell type as ATAC-seq data were used for labeling MPBSs as bound or unbound (Fig 1C). Because of the high false positive rate of TF motifs [6], a large proportion of MPBSs are not really bound by TFs (S1A Fig). To avoid biases due to training and evaluation with a disproportionally large unbound category, we randomly subsampled the bound and unbound MPBSs to have equal proportions for each TF before conforming training, validation, and testing datasets (S1B Fig). The resulting balanced datasets are large enough (more than 5000 labeled MPBSs for most TFs) to enable reliable model training and performance metrics robust to the random subsampling. The trained TAMC models were tested under two types of prediction settings: intra-data prediction (same ATAC-seq data for training and testing) and cross-data prediction (different ATAC-seq data sets for training and testing). In total, we trained and tested TAMC for 47 TFs with published ChIP-seq data for all three cell types in the ENCODE database (S2 Table). Area under the receiver operating curve (AUROC) was used to measure the trained models' performance in classifying bound/unbound MPBSs.

## TAMC outperforms existing methods

To evaluate TAMC performance, we compared TAMC predictions with predictions generated using two representative footprinting tools, namely, TOBIAS and HINT-ATAC. For HINT-A-TAC, we utilized published models that were pre-trained on GM128782 ATAC-seq data for testing [16]. For TOBIAS, it has fixed parameters in its FPS metric and therefore does not require model training before testing [17]. We showed that TAMC gave the best classification of bound and unbound binding sites for most of the 47 TFs under both intra- and cross-data settings (Fig 2A). We further showed that TAMC models trained using multiple (GM12878 and HepG2) ATAC-seq datasets gave better cross-data predictions than models trained using a single (GM12878 or HepG2) ATAC-seq data (Fig 2B). In order to check the influence of sequencing depth on TAMC performance, we downsized the training and testing ATAC-seq datasets to different total numbers of high-quality non-mitochondrial aligned reads. The results showed that higher sequencing depth of the training datasets gave better cross-data classification performance for TAMC (Fig 2C). Compared to TOBIAS and HINT-ATAC, TAMC gave best cross-data classification when the sequencing depth of testing datasets is low; while for testing data with high sequencing depth, TOBIAS gave better classification than TMAC for certain TFs (Figs 2D and S2A). Interestingly, the classification performance of TOBIAS and TAMC were always crossed near the point where the training and testing ATAC-seq datasets have similar sequencing depth

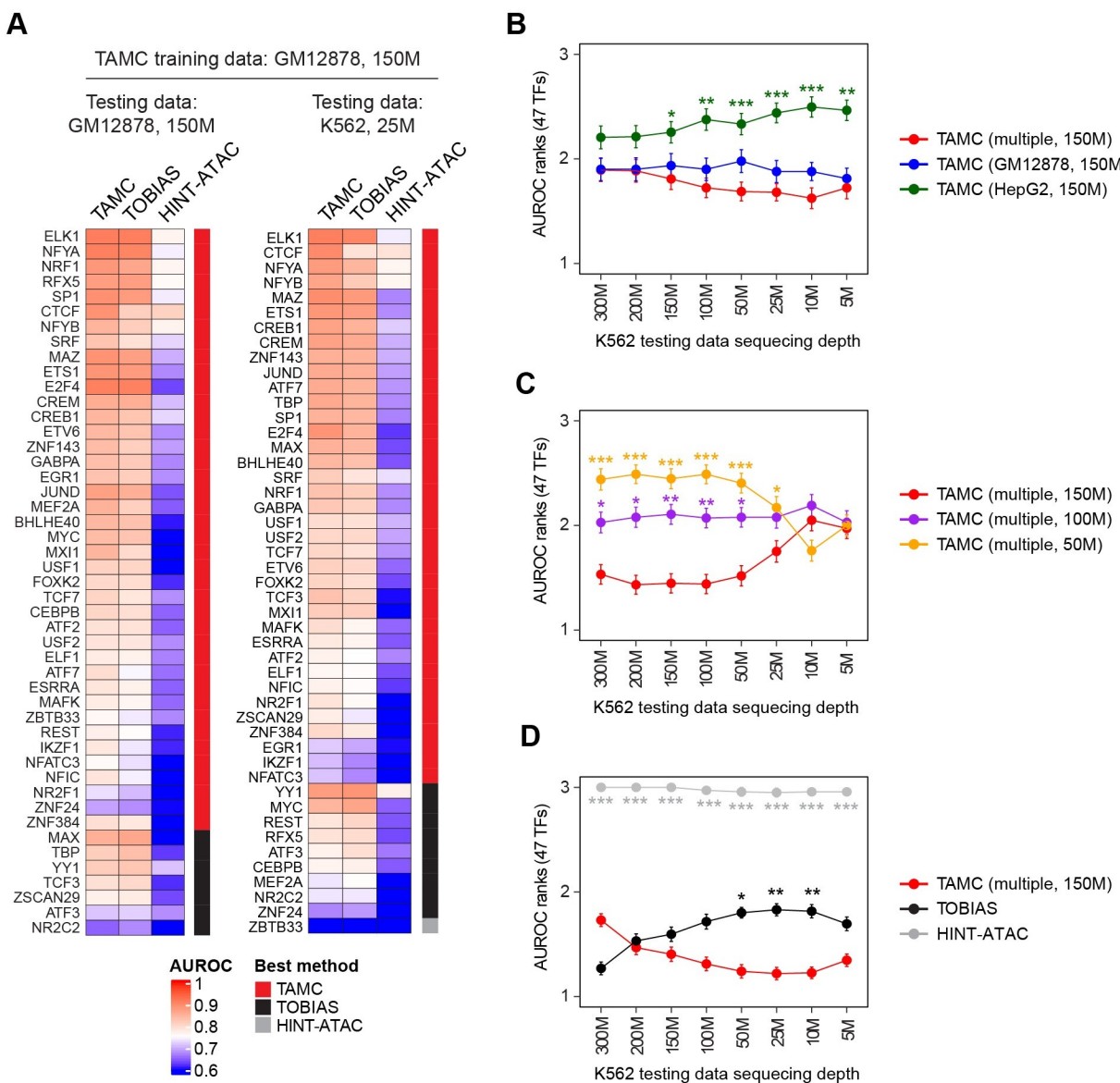

**Fig 2. TAMC outperforms existed methods. (A)** Heat maps compare intra-data (left) and cross-data (right) classification performance of TAMC with TOBIAS and HINT-ATAC. Test data are indicated above each heatmap, and TAMC models were trained using GM12878 ATAC-seq data with 150M sequencing depth in both heatmaps. **(B)** Line graph compares cross-data classification performance of TAMC models trained on ATAC-seq data of GM12878, HepG2 and multiple (GM12878 and HepG2) cell lines. **(C)** Line graph compares cross-data classification performance of TAMC models trained on multiple (GM12878 and HepG2) ATAC-seq data with 150M, 100M and 50M sequencing depth. **(D)** Line graph compares cross-data classification performance of TAMC (multiple, 150M) with TOBIAS and HINT-ATAC. The performance of models within each line graph were ranked from 1 to 3 for each TF. The higher AUROC is, the lower rank number is given. The points in the line graphs show the average AUROC ranks for 47 TFs for each method/model and the error bars represent standard error or mean (SEM). Difference between the performance of TAMC (multiple, 150M) and the other methods/models were examined using Friedman-Nemenyi test and significant differences are labeled ($*$ $p < 0.05$; $**$ $p < 0.01$; $***$ $p < 0.001$). Complete statistic test results for **B**, **C** and **D** are provided in S3 Table. The cell type and sequencing depth of ATAC-seq data used for train TAMC models are indicated within the parenthesis. HINT-ATAC models pre-trained on GM128782 ATAC-seq data were used for all testing, while TOBIAS does not require model training before testing. M denotes million high-quality non-mitochondrial aligned reads.

(Figs 2D and S2B–S2C). These results together suggested that TAMC outperforms TOBIAS and HINT-ATAC in cross-data TFBS prediction as long as the training ATAC-seq datasets were sequenced at higher depth than the testing ATAC-seq datasets.

## TAMC does not require bias correction during input signal processing

Most existing footprinting tools, including TOBIAS and HINT-ATAC, require conducting bias correction during cleavage signal processing to uncover measurable footprint patterns for bound/unbound MPBS classification. However, none of the reported bias correction algorithms can uncover footprint for all TFs faithfully [17]. This makes bias correction a critical step that limits the performance of existing footprint-based methods. To check whether TAMC prediction is affected by bias correction during input signal processing, we compared the classification performance of TAMC models using four different combinations of bias-corrected and uncorrected signals as inputs. Our results showed that the performance of the four TAMC models using uncorrected, partially corrected and fully correct input signals were not significantly different with each other (Fig 3). In addition, all four TAMC models gave better intra-data classification performance than TOBIAS and HINT-ATAC that use bias-corrected signals as optimal inputs (Fig 3). This implies that TAMC models were trained to correct Tn5 cutting bias internally and therefore does not require manual bias correction during input signal processing. Based on these results, we set non-biased corrected input signals as the default input format for TAMC.

## TAMC generates TF-specific models

Both TOBIAS and HINT-ATAC assume that the footprints left by all TFs have the same pattern across the genome–TOBIAS applies the same metric to calculate the footprint scores at all genomic sites and HINT-ATAC uses the model trained for EGR1 to make predictions for all TFs. However, it has been recently reported that the footprint pattern for different TFs are highly heterogeneous from each other [10]. To check whether TAMC can tell the heterogeneity in binding features of different TFs, we compared intra-data prediction performance of TAMC using models trained for the same (intra-TF) or different TFs (cross-TF) as the testing TF (Figs 4 and S4). We found that TAMC give better intra-TF predictions than cross-TF predictions in general (Fig 4). Among the 47 analyzed TFs, 6 TFs (ZNF143, NR2F1, CTCF, MXI1, NFIC and MEF2A) always require TAMC models trained using the same TF for best binding site prediction (Fig 4). In particular, the model trained for CTCF showed extremely high specificity for CTCF binding site prediction (Fig 4). These results suggested that TAMC can tell TF-to-TF differences at their binding sites and generate TF-specific models to improve prediction accuracy.

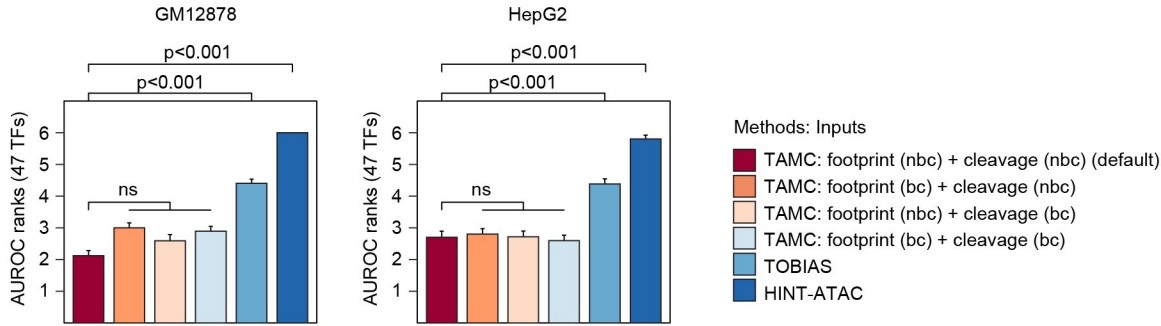

**Fig 3. TAMC performance is independent of bias correction.** Bar graphs compare intra-data classification performance of TAMC models using four different combinations of bias-corrected and uncorrected inputs together with TOBIAS and HINT-ATAC in GM12878 and HepG2 cells. The performance of models within each graph were ranked from 1 to 6 for each TF. The higher AUROC is, the lower the rank number. Average AUROC ranks for 47 TFs for each method/model were shown and the error bars represent SEM. Difference between the performance of default TAMC and the other models were examined using Friedman-Nemenyi test and p-values for significant differences are indicated. Complete statistic test results are provided in S3 Table. bc, bias corrected; nbc, non-bias corrected; ns, non-significant.

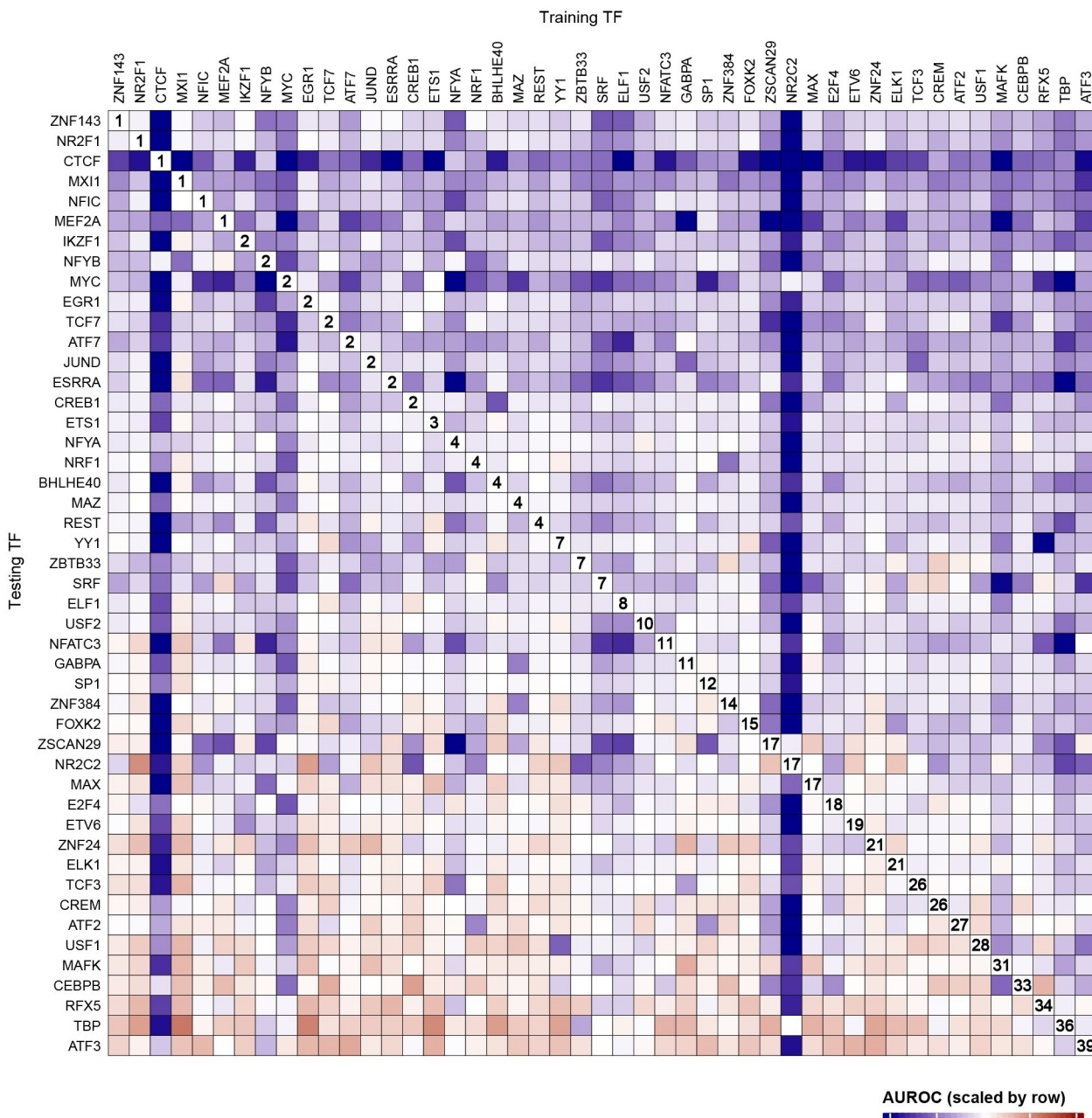

**Fig 4. TAMC generates TF-specific models.** Heat map compares intra-TF and cross-TF performance of TAMC using GM12878 ATAC-seq data. For each testing TF, the prediction performance of different TAMC models was scaled using the formula: scaled AUROC = log₂(AUROC/intra-TF AUROC). Majority of the obtained scaled AUROC values falls within the range from -0.1 to 0.1. Positive scaled AUROC values means better cross-TF predictions than intra-TF predictions, while negative scaled AUROC values means better intra-TF predictions than cross-TF predictions. The rank for intra-TF prediction performance for each testing TF was labeled in the plot. Raw AUROC data are shown in S3 Fig.

## TAMC captures TF-specific binding features by deep learning

To further explore how TAMC exceeds TOBIAS and HINT-ATAC in detecting TFBSs, we tested TAMC with four variant input structures that lack footprint scores (Variant 1), cleavage profile (Variant 2), read strand (Variant 3) and fragment size (Variant 4) information respectively (Fig 5A). By comparing TAMC models using variant 1 and 2 input structures with

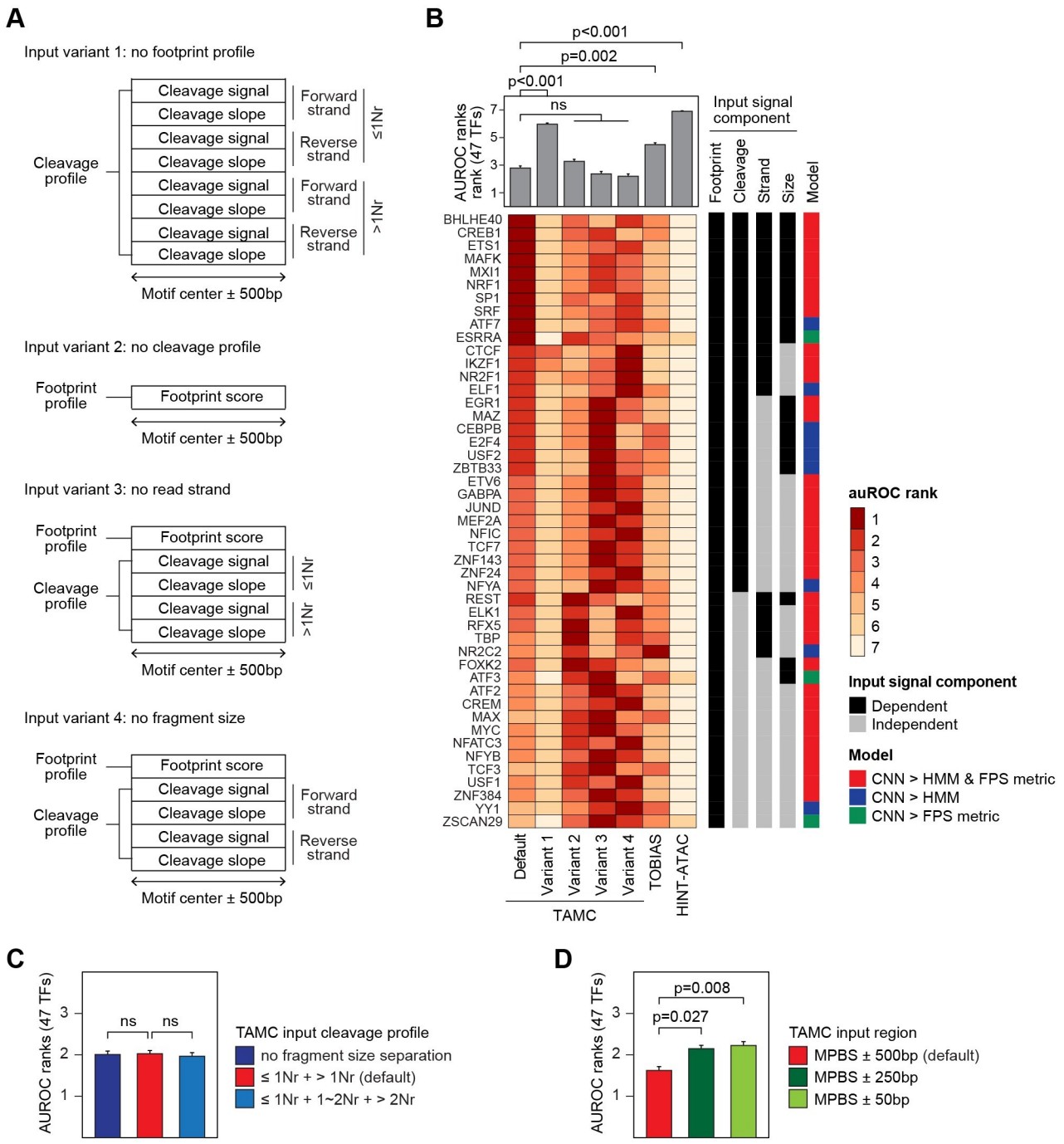

**Fig 5. TAMC captures both footprint and non-footprint features of TFBSs by deep learning. (A)** Schematic of variant TAMC input structures that lack footprint score (Variant 1), cleavage profile (Variant 2), ATAC-seq read strand (Variant 3) and ATAC-seq read fragment size (Variant 4) information, respectively. **(B)** Bar plot and heat map compares intra-data classification performance of TAMC using default and variant input structures together with TOBIAS and HINT-ATAC in GM12878 cells. TFs that require footprint score, cleavage profile, strand or size information within the input signal or the 1D-CNN model for better prediction are indicated in the side bar. **(C)** Bar plot compares intra-data classification performance of TAMC models using inputs with different fragment size separations for the cleavage profile in GM12878 cells. **(D)** Bar plot compares intra-data prediction performance of TAMC models using inputs generated from different region length surrounding MPBSs in GM12878 cells. The models are ranked by their performance within each plot. The higher AUROC is, the lower rank number is given. Average AUROC ranks for 47 TFs for each model were shown in the bar plots (**B**, **C**, and **D**) and the error bars represent SEM. Difference between the performance of default TAMC and the other models were examined using Friedman-Nemenyi test and p-values for selected comparisons are indicated. Complete statistic test results are provided in S3 Table.

HINT-ATAC and TOBIAS respectively, we showed that the 1D-CNN model in TAMC makes better predictions than the classical model (e.g., HMM in HINT-ATAC) or non-model metrics (e.g., FPS metric in TOBIAS) even using the same input signals (Fig 5B). Therefore, the high efficiency in capturing complex and subtle features by 1D-CNN model plays an important role in improving TFBS detection ability of TAMC. At the same time, by comparing performance of default and variant TAMC models, we can define what kind of information in the input signals are important for TAMC to make predictions. Our results showed that TAMC performance was drastically compromised for all 47 TFs when the footprint score profile is removed from input (Fig 5B). In addition, loss of cleavage profiles completely impaired TAMC prediction accuracy for more than half of the TFs (Fig 5B). Therefore, both the footprint and cleavage profiles in the inputs provide important information for TAMC to make predictions.

Interestingly, we determined several TFs requiring information provided by the cleavage profile, including read strand (e.g., CTCF) and/or read fragment size (e.g., EGR1) information, for better binding site prediction (Fig 5B). In consistent with this finding, we detected strand-specific pattern in metagene cleavage profiles at CTCF binding sites (S4 Fig), which can only be captured by TAMC using inputs generated by strand-separated signals. On the other hand, we detected a small footprint pattern at EGR1 binding sites only in chevage profiles generated by reads fragments smaller than one nucleosome size (S4 Fig). Therefore, merging cleavage profiles of small and large reads fragments might dilute the footprint pattern and reduce required information used for EGR1 binding detection. These results suggest that TAMC generates TF-specific models by recognizing TF-specific binding features from ATAC-seq data. In addition, the strand-specific cleavage pattern around CTCF binding site might reflect its dimerized binding mechanisms [22], while the shallow footprint pattern at EGR1 binding sites could reflect its transient activity at most of its binding sites [23]. Therefore, although the overall performance of TAMC was not significantly affected by strand and/or fragment size separation (Fig 5B and 5C), we expected that the performance of TAMC in predicting binding site of specific TFs could be further optimized by adjusting these two factors based on biological mechanisms behind their binding events. Notably, we found that shortening the input regions significantly reduced TAMC performance (Fig 5D). This indicates that TAMC not only captures cleavage features at TFBSs but also surrounding cleavage features that could be affected by binding of co-factors.

## TAMC predicts changes in constitutive CTCF biding sites during ZGA

Our results have revealed that TAMC greatly improved CTCF binding site prediction compared to TOABIS and HINT-ATAC. Studies have shown increased CTCF expression and key roles of CTCF in establishing topologically associating domains (TADs) during human zygotic genome activation (ZGA) [24] (Fig 6A). Because of the limitation in the number and accessibility of human zygotic cells, it is difficult to map CTCF binding sites by ChIP-seq experiment directly. Here we used TAMC to predict CTCF binding sites based on published ATAC-seq data of 2-cell (pre-ZGA) and 8-cell (undergoing ZGA) human embryos [25]. For this prediction, we utilized TAMC models trained on multiple ATAC-seq data (GM12878 and HepG2, sequencing depth = 150 million high-quality non mitochondrial aligned reads), which has been shown to have the best cross-data prediction performance in our study. We predicted sites with binding probability > 0.5 as CTCF binding sties, while sites with binding probability > 0.95 as constitutive CTCF binding sites. As expected, we only detected footprint patten in metagene cleavage profiles at predicted CTCF binding sites but not at the predicted non-CTCF binding sites (S5A Fig). In addition, the predicted constitutive CTCF binding sites exhibited deep footprint patten as a result of persistent TF binding events [23] (S5A Fig).

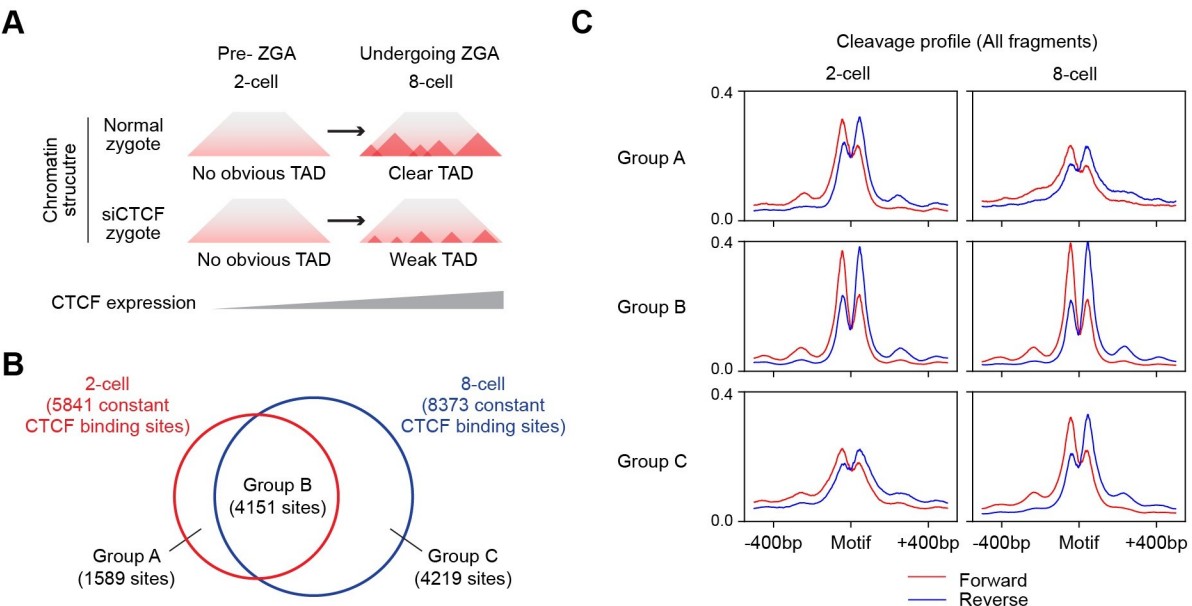

**Fig 6. TAMC predicts changes in constitutive CTCF binding sites during ZGA. (A)** Schematic show CTCF knockdown prevents TAD formation during human ZGA. **(B)** Venn diagram shows the overlap between predicted constitutive CTCF binding sites in 2-cell and 8-cell embryos. **(C)** Metagene plots show aggregated cleavage profiles of predicted 2-cell specific (Group A), common (Group B) and 8-cell specific (Group C) in 2-cell and 8-cell embryos.

These results demonstrated the feasibility of predicting CTCF brining sites in human embryonic cells with TAMC.

We next repeated the prediction experiments for three times and used the common predicted binding sites for the following analyses. We found the number of all CTCF bindings are comparable (less than 5% difference) in 2-cell and 8-cell embryos (S5B Fig). In contrast, there is a drastic (around 40%) increase in the number of constitutive CTCF binding sites at 8-cell stage compared to 2-cell stage (Figs 6B and S6B). In addition, not all constitutive CTCF binding sites in 2-cell zygote are maintained when the embryo enters ZGA. Based on our prediction, 1589 (around 40%) of constitutive CTCF binding sites in 2-cell embryos are lost or start to show compromised CTCF binding activity at 8-cell embryonic stage (Fig 5B and 5C). These results suggest that chromatin structure reorganization during ZGA is associated with both increase in the number and changes in the location of constitutive CTCF binding sites. Constitutively bound CTCF sites have been shown to maintain cell-type specific 3D chromatin architecture in somatic cells [26]. Therefore, the constitutive CTCF binding sites predicted by TAMC in our study could be used as candidate targeting sites for future studies of chromatin structures at specific regions during early human embryonic development.

## Discussion

Predicting binding sites of transcription factors is important for understanding gene expression mechanisms. In this study, we introduced a new tool named TAMC to predict TF binding dynamics at MPBSs using paired-end ATAC-seq data. As summarized in Table 1, TAMC has several advantages comparing to previous tools. First, TAMC does not require bias correction during signal processing, which makes signaling processing easier and avoids further artificial bias caused by the bias correction algorithm. Second, TAMC combines different configurations of processed ATAC-seq signals within its input and therefore retains not only footprint

**Table 1. Summary of features of TAMC and existing footprinting tools for ATAC-seq data.**

| Tools | TAMC | TOBIAS | HINT-ATAC |
|---|---|---|---|
| Year of publication | This paper | 2020 | 2019 |
| Input | Footprint score + Cleavage profile | Footprint score | Cleavage profile |
| Bias correction | Not required | Required | Required |
| Model | 1D-CNN | FPS metric | HMM |
| Model training | Required | Not required | Required |
| Features utilized for TFBS prediction | Footprint + Non-footprint | Footprint | Footprint + Non-footprint |
| TF-specificity | Yes | No | No |
| Sequencing depth | Training > testing | No requirement | No requirement |
| Performance | Intra-data (+++) Cross-data (+++) | (++) | Intra-data (+) Cross-data (+) |

High (+++), medium (++) and low (+) evaluation of TFBS predicting performance.

but also non-footprint information, such as stand and size information, for later prediction use. Finally, TAMC uses 1D-CNN to analyze input signals, which is based on our results, more efficient than classical models (e.g., HMM in HINT-ATAC) or non-model-based metrices (e.g., FPS metric in TOBIAS) in feature capturing. These advantages together allow TAMC to utilize precise binding features of each TF and thus improves its performance in classifying bound/unbound MPBSs.

Most previous studies use the training and testing datasets derived from the same ATAC-seq data for evaluating the tools' performance in classifying bound/unbound MPBSs. However, in real applications, the cell type and sequencing depth of ATAC-seq data for prediction are usually different from available training data. To mimic the situations in real application, we evaluated TAMC performance under both intra-data and cross-data settings. We showed that TAMC outperforms existed methods as long as the training datasets has higher sequencing depth than testing datasets. In real studies, bulk ATAC-seq assays are usually conducted at sequencing depth between 100~200M reads and can provide 20~50M high-quality aligned reads after removing duplicated and mitochondrial DNA. In our study, the TAMC models are trained using ATAC-seq data with 150 million high-quality aligned reads. Therefore, the trained TAMC models generated in this study are suitable for TFBS prediction using most bulk ATAC-seq data generated in real studies. Recently single-cell ATAC-seq is becoming more and more popular because it can analyze multiple cell types at one time. One major obstacle for TFBS prediction using scATAC-seq data is its low sequencing depth for individual cell or specific cell population. As our TAMC models significantly exceeded existing methods in TFBS prediction using ATAC-seq data with low sequencing depth, it could be further applied in analyzing TF binding dynamics using scATAC-seq data in the future.

Among the 47 TFs analyzed in our study, only a small number of TFs (e.g., ATF3 and TBP) did not show better binding site prediction result by TAMC comparing to existed methods (Fig 2A). We noted that these TFs often have very low number of bound MPBSs (S1 Fig), which could result in insufficient data amount for model training. One possible way to improve TAMC's performance for these TFs is to combine labeled MPBSs from different cell types for training. Importantly, our results revealed that in addition to increasing data amount, combining training datasets of multiple cell types can prevent cell type-specific overfitting effect as well, which is important for making cross-data predictions in real applications (Fig 2B). Therefore, while the trained TAMC models generated in our study already outperformed

TOBIAS and HINT-ATAC for most TFs, more robust and generalized models could be obtained by combining deeply sequenced ATAC-seq data of more cell types for training.

Finally, our results showed that ~30% of TFs achieve better performance when using models trained on other TFs compared to intra-TF models (Fig 4). We noted several TFs with low-ranked intra-TF models (e.g., MAX, ELK1 and MAFK) showed higher background noise in their ChIP-seq data than TFs with top-ranked intra-TF models (e.g., CTCF, NR2F1 and EGR1) (Figs 4 and S6). High ChIP-seq background noise is often caused by poor specificity of the antibody used in ChIP experiment. It is unfavorable to ChIP-seq peak summit calling and MPBS labeling, which will then cause the trained model to capture inaccurate binding information. A potential solution is to separate the training into two steps: the first step uses MPBSs of all TFs to train the model to capture common binding features for all TFs (such as the high chromatin accessibility and appearance of footprint pattern), while the second step applies transfer learning by continuing training the model using MPBSs of a specific TF to capture TF-specific binding features (such as the pattern and surrounding chromatin environment of the footprint). For TFs with high noise in their ChIP-seq data and show compromised performance after TF-specific training, we can predict their binding sites just based on the common binding features using the model obtained in the first step.

## Methods

### ATAC-seq and ChIP-seq data processing

ATAC-seq and ChIP-seq data (S1 and S2 Tables) were obtained from the UCSC ENCODE portal (https://www.genome.ucsc.edu/ENCODE). Raw ATAC-seq and ChIP-seq fastq files were trimmed using TrimGalore [27] and aligned with Bowtie2 (v2.3.5.1,—qc-filter—very-sensitive) [28] to reference human genome (h38). Reads with alignment quality lower than 30 or reads aligned to mitochondrial DNA were removed using Samtools (v1.10) [29]. Duplicated reads were removed using MarkDuplicates tool of Picard (v2.0.1; http://broadinstitute.github.io/picard/). The aligned bam files of GM12878 and HepG2 ATAC-seq datasets were further downsized to 150, 100 and 50 million total reads before used for training TAMC models. The aligned bam file for K562 ATAC-seq data was also downsized to 300, 200, 150, 100, 50, 25, 10 and 5 million aligned reads for testing trained TAMC models under different sequencing depth situations. Both ATAC-seq and ChIP-seq peaks regions and summits were called using MACS2 (v2.2.7.1,—nomodel—nolambda—keep-dup auto—call-summits) [30].

### MPBS labeling and sampling

Binding motifs for 47 TFs were all obtained from JASPAR CORE 2020 database (https://jaspar.genereg.net/). For TFs with redundant motifs, only the latest version of motifs was used. MPBSs for each TF across hg38 genome were mapped using the MOODS with a p-value threshold of 0.0001 [4]. Only MPBSs located within open chromatin regions (ATAC-seq peak regions) were used for further analyses. To label the TF-bound and unbound status of each MPBS, ChIP-seq data obtained from the same cell type as ATAC-seq data were used. MPBSs located outside ChIP-seq peak regions were labeled as unbound sites. For MPBSs located within each ChIP-seq peak region, only the MPBS located closest to and within 50bp from the highest ChIP-seq summit within that peak was kept and labeled as TF-bound in later analyses. If one TF has more unbound MPBSs than bound MPBSs, the obtained unbound MPBSs will be further randomly sampled to the same number as bound MPBSs for further analyses and vice versa.

For GM12878 and HepG2 cells, 70% of labeled MPBSs were used for modeling training, 20% of labeled MPBSs were used for validation during training, and the remaining 10% of

labeled MPBSs were used for testing the trained models. Equal number of bound and unbound MPBSs for each TF were randomly sampled into the training, validating and testing datasets. For K562 cells, all labeled MPBSs were used to prepare testing input signals.

## Input signal processing

To prepare input signals for TAMC, the ATAC-seq data was processed using ATACorrect and FootprintScores tools in TOBIAS package to generate footprint scores at single-based resolution [17]. Footprint scores within 500bp form the center of each MPBS was made into a 1x1000 footprint feature vector. At the same time, the ATAC-seq data were separated into 4 files by strand and fragment sizes, and then each file was used for counting genomic cleavage signals and calculating slopes of cleavage signals within 500bp from each MPBS center following the signal processing steps in HINT-ATAC package [16]. The obtained 4-channels of genomic cleavage signals and 4 channels of cleavage slopes were combined into 8x1000 cleavage feature vectors. The default TAMC input signal was generated by concatenating the 1x1000 footprint feature vector and the 8x1000 cleavage feature vector to form the 9x1000 input feature vector. In addition, the 1x1000 footprint feature vector and the 8x1000 cleavage feature vector form two variant TAMC input structures lacking cleavage profiles and footprint information by themselves. To make the 5x1000 variant input feature vectors lacking size or strand information, the 1x1000 footprint feature vector was concatenated with a 4x1000 cleavage feature vector generated using ATAC-seq data only separated by strand or fragment sizes respectively.

## Training

The TAMC model was trained using input signals generated using ATAC-seq data of GM12878 and/or HepG2 cells. The python package PyTorch [31] was used for generating and training the model (Fig 1A). The model is trained using the Adam algorithm with minibatch size of 35, epoch size of 10 and maximum iteration number of 100. The datasets were randomized between each epoch of training. Validation loss is evaluated at the end of each training iteration and trained models with lowest 3 validation losses were saved as best models.

## Testing

The trained TAMC models were tested sing ATAC-seq data of GM12878, HepG2 or K562 cells. For each TF, the AUROC value was calculated based performance in classifying bound/unbound MPBS using the average of binding probabilities predicted by the 3 best models.

The performance of TAMC was compared to TOBIAS and HINT-ATAC. TOBIAS does not require model training before testing, and its AUROC values were calculated based on the ability of the average footprint scores within each MPBS to classify bound/unbound sites. To test HINT-ATAC performance, we used the published model trained on EGR1 using GM12878 ATAC-seq data. To calculate AUROC values for HINT-ATAC prediction results, all MPBSs were first ranked by the PWM scores generated by MOODS and the maximum PWM score was saved. Then footprint regions were identified using the footprinting tool of rgt-hint package [16]. MPBSs overlapping footprint regions were assumed to be bound and their PWM scores were renewed by adding the maximum PWM score. The other MPBSs were regarded as unbound and their PWM scores were kept unchanged. AUROC is calculated based on the ability of the final PWM scores in classifying bound/unbound MPBSs.

For each TAMC model, the experiments (including MPBS sampling, training, and testing) were repeated for three times. The performance of TOBIAS and HINT-ATAC were also tested for three times using the testing MPBSs generated in each experiment replicate. Raw AUROC

values were plotted into heat maps (Figs 2A and S3) or directly provided in S4 Table. For each testing TF, AUROC values generated by different methods/models were first ranked in each experiment; then the mean AUROC rank of the three experimental replicates were used for comparison.

## Prediction

To predict CTCF binding sites during ZGA, we first downsized ATAC-seq data of 2-cell and 8-cell embryos to the same sequencing depth at 40,000,000 high-quality non-mitochondrial aligned reads. ATAC peaks were called as mentioned above, and ATAC peaks of these two cell types were merged as common open chromatin regions. We then selected CTCF MPBSs located within common open chromatin regions for later TAMC prediction. The prediction was repeated for three times using the three sets of models obtained from three training experiments mentioned above. Predicted binding probabilities at all CTCF MPBSs in three experiments are provided in S5 Table. CTCF MPBSs with predicted binding probability larger than 0.5 were regarded as CTCF binding sites. CTCF MPBSs with predicted binding probability larger than 0.95 were regarded as constitutive CTCF binding sites.

## Supporting information

**S1 Fig. Number of labeled MPBSs for 47 TFs before (A) and after (B) equalization of bound and unbound sites in GM12878 cells (supplementary to Figs 1 and 2).** TFs did not show better intra-data classification using TAMC than TOBIAS or HINT-ATAC were highlighted in red.
(TIF)

**S2 Fig. TAMC performance is affected by sequencing depth of training data (supplementary to Fig 2).** (A) Line graphs compare CTCF, SRF, E2F4 and NFIC binding site prediction performance using trained TAMC model with TOBIAS and HINT-ATAC. The points represent for mean AUROC ± SEM (n = 3 experimental replicates). P-values were calculated using two-sided unpaired Student's t-test. (B-C) Line graphs compare average cross-data classification performance of TAMC models trained on multiple (GM12878 and hepG2) ATAC-seq data with 100M (A) and 50M (B) sequencing depth with TOBIAS and HINT-ATAC. The performance of models within each line graph were ranked from 1 to 3 for each TF. The higher AUROC is, the lower rank number is given. The points show the average AUROC ranks for 47 TFs for each method and the error bars represent SEM. P-values were calculated using Friedman-Nemenyi test. The cell type and sequencing depth of ATAC-seq used for train TAMC models are indicated within the parenthesis. HINT-ATAC models pre-trained on GM128782 ATAC-seq data were used for all testing, while TOBIAS does not require model training before testing. M denotes million high-quality non-mitochondrial aligned reads. Complete statistic test results and raw AUROC data for all figures are provided in S3 and S4 Tables, respectively. * $p < 0.05$, ** $p < 0.01$, *** $p < 0.001$.
(TIF)

**S3 Fig. Heatmap shows intra-TF and cross-TF classification performance (AUROC) of TAMC using GM12878 ATAC-seq data (supplementary to Fig 4).**
(TIF)

**S4 Fig. Metagene plots show aggregated cleavage profiles within ±500bp regions of labeled CTCF and EGF1 MPBSs in GM12878 cell.** Cleavage signals were processed without manual bias correction. Three plots were made for each class of MPBSs using all ATAC-seq reads, ATAC-seq reads with maximum fragment size at 1Nr, and ATAC-seq reads with minimum

fragment size more than 1Nr. Nr, nucleosome size. MPBS, motif-predicted binding site.
(TIF)

**S5 Fig. Predicted CTCF binding sites in human embryonic cells. (A)** Metagene plots show aggregated cleavage profiles at predicted unbound, bound, and constitutively bound CTCF MPBSs in 8-cell embryos. **(B)** Venn diagrams show the number of predicted bound/conductively bound CTCF MPBSs in 2-cell and 8-cell embryos from 3 experiments. CTCF binding sites predicted in all three experiments are regarded as true binding sites.
(TIF)

**S6 Fig. Representative genome tracks show ChIP-seq signals of TFs with top-ranked intra-TF model (CTCF, NR2F1 and EGR1) and low-ranked intra-TF model (MAX, ELK1 and MAFK).**
(TIF)

**S1 Table. List of ATAC-seq datasets.**
(XLSX)

**S2 Table. List of TF motifs and ChIP-seq datasets.**
(XLSX)

**S3 Table. Complete results of significance tests.**
(XLSX)

**S4 Table. Raw AUROC values for evaluation of different models and methods.**
(XLSX)

**S5 Table. Raw predicted binding probabilities for imputing CTCF binding sites in human embryos.**
(XLSX)

## Acknowledgments

We thank Dr. Lawrence Carin for providing suggestions for this project and Shelley Rusincovitch for organizing the Duke Data Science Plus (+DS) program. For project involving non-sensitive data, +DS is supported by Duke Research Computing for the use of the Duke Compute Cluster for high-throughput computation. The Duke Data Commons storage is supported by the National Institutes of Health (1S10OD018164-01).

## Author Contributions

**Conceptualization:** Tianqi Yang, Ricardo Henao.

**Data curation:** Tianqi Yang.

**Formal analysis:** Tianqi Yang.

**Investigation:** Tianqi Yang.

**Methodology:** Tianqi Yang.

**Project administration:** Ricardo Henao.

**Software:** Tianqi Yang.

**Supervision:** Ricardo Henao.

**Validation:** Tianqi Yang.

**Visualization:** Tianqi Yang.

**Writing – original draft:** Tianqi Yang.

**Writing – review & editing:** Tianqi Yang, Ricardo Henao.

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
