## [Decision Letter · Decision Letter 0]

19 Apr 2022

Dear Yang,

Thank you very much for submitting your manuscript "TAMC: A deep-learning approach to predict motif-centric transcriptional factor binding activity based on ATAC-seq profile" for consideration at PLOS Computational Biology.

As with all papers reviewed by the journal, your manuscript was reviewed by members of the editorial board and by several independent reviewers. In light of the reviews (below this email), we would like to invite the resubmission of a significantly-revised version that takes into account the reviewers' comments.

We cannot make any decision about publication until we have seen the revised manuscript and your response to the reviewers' comments. Your revised manuscript is also likely to be sent to reviewers for further evaluation.

Sincerely,

Saurabh Sinha

Guest Editor

PLOS Computational Biology

Ilya Ioshikhes

Deputy Editor

PLOS Computational Biology

Reviewer's Responses to Questions

**Comments to the Authors:**

Reviewer #1: The paper describes an interesting deep-learning approach to predict

transcription factor binding sites using motif information, ATAC-seq

profiles, and ChIP-seq information. While it seems a worthwhile

approach, I feel some aspects would benefit from greater clarity.

Major:

Results, page 6, "genomic cleavage signals and slopes are generated

following HINT-ATAC input signal processing strategy..." is the

strategy reimplemented, or is HINT-ATAC used directly?

page 7: "we compared TAMC predictions with predictions generated using

previous footprinting tools including TOBIAS and HINT-ATAC". These are

the only two previous tools compared. The sentence should be

reworded.

page 8, "compared to TOBIAS and HINT-ATAC, TAMC gave best cross-cell

prediction..." however, table 1 suggests that the former two tools

cannot be used for cross-data. This should be clarified.

page 12, figure 4: Why are the AUC values scaled between -1 and 1?

This seems an unusual procedure and makes it impossible to compare

roews. Why not plot the raw AUC on a 0-1 scale as usual? This should

be clarified

General: a bit more detail in the description of the deep learning

architecture would be desirable.

General: It would be good to include error bars wherever possible, ie

figures 2b, 2c, 3.

Minor:

Abstract: "TAMC captures both footprint and non-footprint features..."

do the authors mean "utilizes" (not "captures")?

Author summary: avoid acronyms here, like CNN, TFBS, ATAC-seq, or

explain them, since this is for non-specialists

Reviewer #2: Review of TAMC: A deep-learning approach to predict motif-centric transcriptional factor binding activity based on ATAC-seq profile

In this paper, the authors introduced a new computational method, TAMC, predicting TF-binding activity in open chromatin regions from ATAC-seq data. The authors showed that TAMC outperformed two existing methods, TOBIAS and HINT-ATAC, by combining them using a deep-learning approach without the correction of sequence bias on cut sites (e.g., Tn5 sequence preference). However, it is unclear how the convolutional network structure of TAMC is configured to train the dependency of local TF footprints and cleavage signals reflecting TF-binding biochemistry. It is also unclear whether TAMC can be generalized to other data sets. These points are discussed in more detail below:

Major points:

1) The convolutional layer architecture of TMAC shown in Figure 1A does not seem to integrate neighboring base-pair signals within a specific feature. Instead, it aggregates signals across neighboring features at the same position. In other words, the current architecture seems to randomly convolute footprinting and cleavage site features, which are then max-pooled without considering neighboring base pair signals. Since the order of the features is arbitrary, I am struggling to understand how the convolutional layer learns the biology of transcriptional binding sites. What do three different convolutional layers with kernel sizes (k=3, 5, 7) and color (red, green, blue) mean from a biological perspective?

2) It is also unclear why such a long region (1,000 bp) is needed for modeling. A footprinting event is local and can be captured within 50bp for almost all TFs. Is it really required to have 1,000bp to achieve high accuracy? What happens if it is reduced to 500bp or even 100bp? What is the optimal length?

3) When training and testing the model of TAMC, the authors used only three cell lines (GM12878, HepG2, and K562), for which many ChIP-seq datasets are available in the ENCODE database. However, no such ChIP-seq data sets are available for most human tissues, and it is unclear if the model can be extended to these human tissues. The authors need to demonstrate the generalizability of the model by analyzing bulk/single-cell ATAC-seq data sets from human tissues.

4) In Figure 5B, the authors primarily used CTCF and EGR to demonstrate that TAMC can capture footprint features of TF-binding sites. However, a more systematic analysis is required to thoroughly test if the differential prediction performance across different model structures is specific to TFs. For example, the expression and the size of TFs could be associated with differences in the prediction performance across the model variants. This can potentially be used to improve the prediction performance further.

5) The authors showed that TAMC achieved robust performance regardless of the bias correction, implying that the TAMC model also learned features associated with bias and corrected them internally. If this is the case, I expect the prediction results to be concordant between the bias-corrected and uncorrected inputs. Please demonstrate this.

6) Direct and quantitative comparison of AUC scores is required throughout the manuscript. In Figure 2A, it is unclear how significantly AUCs are different between TAMC and TOBIAS/HINT-ATAC. The distribution of AUC scores should be compared among the three methods, and the statistical significance of the differences should be assessed. The average AUC rank is not a good metric, either. Please consider it to something more direct in Figures 2B/C, 3, and 5B. In Figure 4, what is the normalization method to scale AUC (-0.1~0.1) by row? It is also unclear why the AUC should be scaled in this analysis.

Minor points:

1) AUC is the abbreviation of Area Under Curve, and it is not ROC-curve specific. Please mention this is the AUC of ROC in the method section.

2) MPBS is not a commonly used acronym. Please consider not using it.

3) Typos:

• “stand” in line 12 of page 13 should be “strand.”

• “sing” in line 6 of page 19 should be “using.”

Reviewer #3: This paper proposes a CNN model to predict TF binding site (TFBS) using footprint and non-footprint (cleavage signal and slope) features derived from ATAC-seq data. Their TF-specific CNN model aggregates the footprint scores at base resolution and cleavage profiles using 1-D convolutional filters in the region (1000 bps) around the center of the motif predicted binding sites. It is shown that the model performance is independent of correcting for the sequence-specific cleavage bias present in the ATAC-seq data; a property that does not exist in the other tools for TFBS prediction using ATAC-seq profiles.

I have the following questions,

1) Please explain why the cutoff of 1 nucleosome size was used for the ATAC-seq reads (Figure 1B)? Also, explain why adding more intervals for the read size might be redundant, e.g., < 1Nr, 1Nr < - < 3 Nr, >3Nr, given that the inclusion of read size information affects the performance (Figure 5B)?

2) Please explain the choice of K562 cell line as the target dataset and GM12878 and HepG2 as the source datasets in the cross-data experiments (Figure 2B,C).

3) In Figure 2C, which dataset(s) were used as the source dataset for TOBIAS and HINT-ATAC?

4) Average AUC rank over the TFs is used to compare the performance of different models or training settings. Please provide a way for testing/explaining the significance of delta X in average AUC rank, i.e., is a decay of 0.1 in average AUC rank significant?

**Have the authors made all data and (if applicable) computational code underlying the findings in their manuscript fully available?**

Reviewer #1: Yes

Reviewer #2: Yes

Reviewer #3: Yes

PLOS authors have the option to publish the peer review history of their article (what does this mean?). If published, this will include your full peer review and any attached files.

Reviewer #1: **Yes: **Rahul Siddharthan

Reviewer #2: No

Reviewer #3: **Yes: **Saba Ghaffari
---

## [Decision Letter · Decision Letter 1]

26 Jul 2022

Dear Yang,

Thank you very much for submitting your manuscript "TAMC: A deep-learning approach to predict motif-centric transcriptional factor binding activity based on ATAC-seq profile" for consideration at PLOS Computational Biology. As with all papers reviewed by the journal, your manuscript was reviewed by members of the editorial board and by several independent reviewers. The reviewers appreciated the attention to an important topic. Based on the reviews, we are likely to accept this manuscript for publication, providing that you modify the manuscript according to the review recommendations.

While Reviewers 1 and 3 are now fully satisfied with the manuscript, Reviewer 2 has minor concerns about the revision. Please address these to the extent possible.

Sincerely,

Saurabh Sinha

Guest Editor

PLOS Computational Biology

Ilya Ioshikhes

Deputy Editor

PLOS Computational Biology

[LINK]

While Reviewers 1 and 3 are now fully satisfied with the manuscript, Reviewer 2 has minor concerns about the revision. Please address these to the extent possible.

Reviewer's Responses to Questions

**Comments to the Authors:**

Reviewer #1: I find the manuscript greatly improved and the responses to all reviewers thorough and clear.

Two minor points:

* "including" implies a non-exhaustive list ("two representative footprinting tools including TOBIAS and HINT-ATAC"). I would suggest "namely", ie "two representative footprinting tools, namely, TOBIAS..."

* "matric" (multiple places) should be "metric" I think. Eg, "Tobias applies the same matric...", "FPS matric" etc.

Reviewer #2: The authors addressed most of my concerns. The revised Figure 1A makes it much easier to understand the CNN structure used in this study. However, I still have some questions:

1) I am still trying to understand why the model requires longer input regions to achieve the best performance. I am wondering if there is a positional bias between bound and unbound MPBSs. For example, bound MPBSs may be more centrally positioned with respect to ATAC-seq peak summits, while unbound MPBSs are more peripheral, which may introduce systematic ATAC-seq signal bias. If this is the case, the difference in ATAC-seq read intensity in the 1kb region centered at MPBS may be enough to distinguish unbound MPBS from bound MPBS. Please demonstrate that there is no positional bias. If the bias exists, the authors should control it when making the balanced training set when subsampling the unbound MPBSs, and repeat the analysis.

2) The raw AUROC for all experiments is quite helpful to better understand the results. As such, I would suggest that the authors directly compare AUROCs across different models using 2~3 representative TFs.

3) After a better understanding of “scaled AUC” in Figure 4, I find it quite interesting that ~30% of TFs achieve better performance when using models trained on other TFs. Although I agree with the authors that intra-TF prediction is generally better than cross-TF prediction for most TFs, I am also curious if there are any common properties shared by these “unusual” TFs.

Reviewer #3: All of my questions have been answered to my satisfaction.

**Have the authors made all data and (if applicable) computational code underlying the findings in their manuscript fully available?**

Reviewer #1: Yes

Reviewer #2: Yes

Reviewer #3: Yes

PLOS authors have the option to publish the peer review history of their article (what does this mean?). If published, this will include your full peer review and any attached files.

Reviewer #1: No

Reviewer #2: No

Reviewer #3: No

Figure Files:

Data Requirements:

Reproducibility:

References:

---

## [Editor Report · Decision Letter 2]

24 Aug 2022

Dear Yang,

We are pleased to inform you that your manuscript 'TAMC: A deep-learning approach to predict motif-centric transcriptional factor binding activity based on ATAC-seq profile' has been provisionally accepted for publication in PLOS Computational Biology.

Best regards,

Saurabh Sinha

Guest Editor

PLOS Computational Biology

Ilya Ioshikhes

Section Editor

PLOS Computational Biology

---

## [Editor Report · Acceptance letter]

8 Sep 2022

PCOMPBIOL-D-22-00225R2 

TAMC: A deep-learning approach to predict motif-centric transcriptional factor binding activity based on ATAC-seq profile

Dear Dr Yang,

I am pleased to inform you that your manuscript has been formally accepted for publication in PLOS Computational Biology. Your manuscript is now with our production department and you will be notified of the publication date in due course.

With kind regards,

Anita Estes
